# Spatiotemporal Variability of Intensity–Duration–Frequency (IDF) Curves in Arid Areas: Wadi AL-Lith, Saudi Arabia as a Case Study

Ibrahim H. Elsebaie [1], Mohamed El Alfy [2,3] and Atef Qasem Kawara [1,*]

1  Civil Engineering Department, College of Engineering, King Saud University, Riyadh 11421, Saudi Arabia; elsebaie@ksu.edu.sa
2  Prince Sultan Institute for Environmental, Water, and Desert Research, King Saud University, Riyadh 11421, Saudi Arabia; alfy@mans.edu.eg
3  Geology Department, Faculty of Science, Mansoura University, Mansoura 35511, Egypt
*  Correspondence: 439106883@student.ksu.edu.sa

**Abstract:** In arid areas, flashflood water management is a major concern due to arid climate ambiguity. The examining and derivation of intensity–duration–frequency (IDF) curves in an urban arid area under a variety of terrain patterns and climatic changes is anticipated. Several flood events have been reported in the Al-Lith region of western Saudi Arabia that took away many lives and caused disruption in services and trade. To find and examine the extremities and IDF curves, daily rainfall data from 1966 to 2018 is used. The IDF curves are created for a variety of return periods and climate scenarios in three terrain variabilities. This research examines various distributions to estimate the maximum rainfall for several metrological stations with varying return periods and terrain conditions. Three main zones are identified based on ground elevation variability and IDF distributions from upstream in the eastern mountainous area to downstream in the western coastal area. These IDF curves can be used to identify vulnerable hotspot areas in arid areas such as the Wadi AL-Lith, and flood mitigation steps can be suggested to minimize flood risk.

**Keywords:** rainfall intensity; intensity–duration–frequency (IDF) curves; arid areas; terrain variability; Saudi Arabia

## 1. Introduction

Flash floods are one of the most common environmental challenges in arid regions, threatening the protection of scarce natural resources. Precipitation patterns and extremes in different parts and seasons are highly variable in arid areas [1]. The response of drainage systems to projected climate changes is site dependent [2,3]. There is an increased attention on urban stormwater management and flood mitigation in vulnerable arid regions [4]. Recent extreme rainfall events resulting in flooding took away many lives and caused disruption in services and trade, including transport and communication. During the flood in 2018 which was of high intensity, residential areas in the Al-Lith area of west Saudi Arabia were submerged in rainwater. As a result, there is a greater emphasis on urban storm water management and flood mitigation. Therefore, policymakers have been compelled to take steps to address the effect of climate change on local conditions, as well as adaptation and mitigation actions that can be integrated into decision-making processes.

The IDF curves that define the relation between rainfall intensity, rainfall duration, and return period are commonly used in the design of hydrologic, hydraulic, and water resource systems. The IDF curves are focused on stormwater management and engineering hydraulic infrastructure design in order to withstand dangerous floods [5–9]. The IDF curve gives the expected rainfall intensity of a given duration of storm having desired frequency of occurrence. It describes designed storms that are based on the average characteristics of

storms that existed in the past. Hypothetical storms with average characteristics of storm events can expect extreme precipitation and flood events in the future.

Several empirical functions and generalized IDF equations were developed for the different regions [10–12]. To build a mathematical system of IDF curves, an efficient parameterization technique was used. IDF curves for ungauged sites in the eastern United States were updated using rainfall level techniques and iso-pluvial maps [13]. In the Sinai Peninsula in the northeastern part of Egypt, regional IDE curves from isolation maps for untapped sites were derived [14]. IDF curves are applied to ungauged locations using weather station records that have been adjusted for deviations within the standard range [15].

By applying suitable statistical distributions based on the rainfall records such as Gumbel and log-Pearson type III distributions, IDF curves can be created. Better estimates of rainfall depth and intensity can be achieved by making long-term records available to enhance recorded storm intensity [16]. When nothing in the series suggests whether one distribution is more likely to be suitable than another [17], the optimal distribution is selected using the goodness-of-fit test, which measures how well the evaluated distributions match the given data [18]. The Kolmogorov–Smirnov (K–S) and Chi-square tests help in choosing the best distribution [19]. The root mean square error is used to compare various probability distributions and determine which is better for the data over a 24 h period and then for the subintervals [9]. Terrain variability is a significant factor, with little variation in outcomes between the various IDF approaches in the Riyadh area, where flat topography and an arid climate prevail [20]. Regional maps of likely maximum precipitation and IDF curves were created in order to predict flood incidence in the Al-Madina area north of Al-Lith to estimate discharge when building flood control systems [21,22]. Abdeen et al. [23] looked at the distribution of maximum daily rainfall in Saudi Arabia and found that the log-Pearson Type III (LPT III) distribution was the best model. Al-Areeq et al. [24] used Gumbel and LPT III distribution to develop IDF curves for five areas (Abha, Al-Baha, Bisha, Gizan, and Khamis Mushait) in the southwestern region of Saudi Arabia. The authors of [25] developed IDF curves for Krishna District, India, using log-normal, normal, and Gumbel distributions.

Koutsoyiannis and Baloutsos [26] investigated a 136-year series of maximum daily rainfall in Athens. The statistical analysis of these data show that the employed extreme value type I (EV1 or Gumbel) distribution is inappropriate for the examined record (especially in its upper tail); this distribution would appear to be an appropriate model if fewer years of measurements were available, whereas the general extreme value (GEV) distribution appears to be suitable. Kastridis and Stathis [27] analyzed the annual maximum daily rainfall of three sites in Greece and observed that the Generalized Extreme Value (GEV) and Extreme Value Type I (EV-1) distributions are the best fit for the data rainfall available.

The aim of this research was to examine various distributions with the variability of rainfall and terrain characteristics with varying return periods and terrain conditions. The study provides insights into how climate and terrain variability can be incorporated into local planning and decision making through an integrated approach for sustainable urban water management. Values of rainfall intensity at a specific rainfall duration are usually extracted from the IDF curves and utilized for the design of hydraulic structures used for flood protection and urban drainage systems. Therefore, it is very important to estimate the design rainfall intensity as accurately as possible.

## 2. Study Area

Wadi Al-Lith area is bounded by the high mountains of the As-Sarawat from the north and east and the Red Sea coast from the west (Figure 1). The Wadi Al-Lith covers an area of 3089 km$^2$ and an estimated length of 109 km. It is located between longitudes 40°11′26″ and 40°48′44″ and between latitudes 20°7′54″ and 21°7′7″. Al-Laith city is a moderate city located downstream of Wadi Al-Lith, 180 km to the southwest from the holy city of Mecca in the western part of Saudi Arabia, where several international roads pass

by the study area. Saudi Arabia is classified as arid to hyper-arid country. The study area has its maximum temperatures in July (40.1 °C), whereas the coldest months are January and February (29.6 °C). The lowest monthly average value of relative humidity occurs during April to July (<20%), while the highest monthly average value occurs in January and December (>90%). The variation in relative humidity shows a trend to differences in elevation and the effects of the escarpment ridge. The highest monthly average of potential evaporation is in July (200 mm), while the lowest value is in February (111 mm). The general wind direction in the study area is west to southwest in summer and northwest to west in winter. The maximum monthly average wind speed occurs during September (39 km/h) while its minimum occurs during December (17.3 km/h).

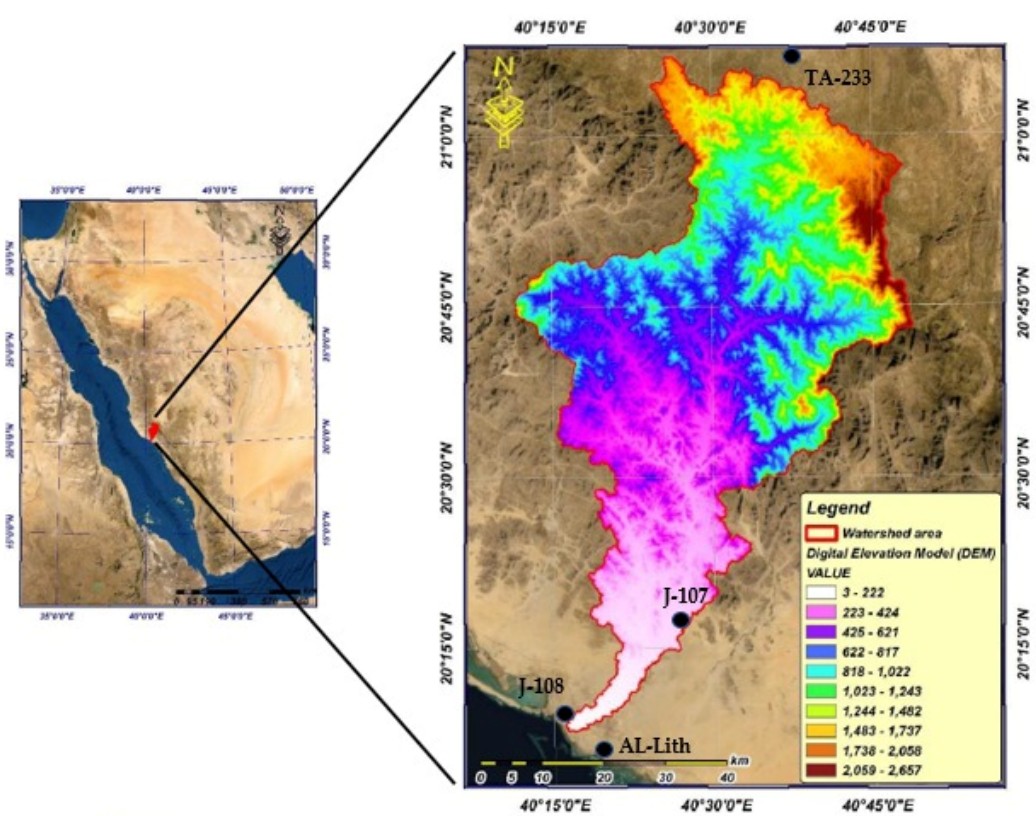

**Figure 1.** General location map and digital elevation model (DEM) of Wadi Al-Lith.

The Wadi Al-Lith catchment area is an integral part of the Arabian Shield; the area is mainly covered by metavolcanic, metasediments and plutonic rocks, and Quaternary alluvial deposits. The oldest rocks are composed of metavolcanic and related rocks of the Baish group and interlayered metasedimentary rocks of the Sadiyah formation. The Sadiyah formation, Baish group, and Lith suite were intruded by granodiorite rocks of the Jumah suite and by granitic rocks of the Hurqufah suite; both suites are late Proterozoic [28,29]. During the Tertiary, subalkaline and alkaline igneous rocks were intruded and extruded over a long period to form the Damm dike complex and the Sita formation. These rocks are bounded by faults that are parallel to the margin of the Red Sea. Major structural features in Al-Lith area are related to Precambrian northwest–southeast to east–west compression and to Tertiary northeast–southwest tension.

## 3. Methodology and Data Collection

The best fit of certain probability distributions can be found by analyzing IDF curves of extreme rainfall quantities of fixed length. IDF curves are derived from frequency analyses of rainfall measurements. Based on rainfall measurements for any recorded year, the annual maximum rainfall intensity for different durations is estimated. Common

durations for design applications are 5-min, 10-min, 15-min, 30-min, 1 h, 2 h, 6 h, 12 h, and 24 h [16]. The IDF curves are generated by calculating each series of annual maxima for each frequency; one for each period of rain. Every frequency analysis has the prime objective of determining the function of the rain intensity exceedance probability distribution for each period. Rainfall data was gathered from the Ministry of Agriculture, Water, and Environment's data base. Regular rainfall depth for J107 and J108 stations was collected from 1966 to 2018, as well as rainfall depth for TA23 station from 1971 to 2005. The Gumbel distribution and the log-Pearson type III distribution are applied and compared with select the best distribution that is suitable for the data set. Estimated rainfall in mm and its intensity in mm/hr were calculated for different return periods and durations.

Gumbel [30] is the commonly used distribution for IDF analysis, the Gumbel distribution frequency precipitation $P_T$ (in mm) for each duration with specified return period $T$ (in year) is given by Equation (1):

$$P_r = P_{ave} + KS \tag{1}$$

where $K$ is the Gumbel frequency factor given by:

$$K = -\frac{\sqrt{6}}{\pi}\left[0.5772 + ln\left[ln\left[\frac{T}{T-1}\right]\right]\right] \tag{2}$$

where $P_{ave}$ is the average of the maximum precipitation corresponding to a specific duration.

In utilizing Gumbel's distribution, the arithmetic average in Equation (1) is used:

$$P_{ave} = \frac{1}{n}\sum_{i=1}^{n} p_i \tag{3}$$

where $p_i$ is the individual extreme value of rainfall and $n$ is the number of events or years of record. The standard deviation is calculated by Equation (4) computed using the following relation:

$$S = \left[\frac{1}{n-1}\sum_{i=1}^{n} (p_i - p_{ave})^2\right]^{\frac{1}{2}} \tag{4}$$

where $S$ is the standard deviation of $P$ data. The frequency factor ($K$), which is a function of the return period and sample size, when multiplied by the standard deviation gives the departure of a desired return period rainfall from the average. Then the rainfall intensity, $I$ (in mm/hr) for return period $T$ is obtained from:

$$I_t = \frac{P_T}{T_d} \tag{5}$$

where $T_d$ is durations in hours.

The annual maximum sequence, which consists of the highest values found in each year, is commonly used to describe the frequency of rainfall. The peak-over-threshold definition, which consists of all precipitation quantities above certain thresholds chosen for different durations, is an alternative data format for rainfall frequency studies. The annual-maximum-series approach is more widely used in practice due to its simpler structure [31]. The simplified expression for this latter distribution is given below for the frequency precipitation obtained using log-Pearson type III. The log-Pearson III technique is obtained in the same manner as the Gumbel technique [32] but using the logarithm of variables as in the relation:

$$P^* = Log(p_i) \tag{6}$$

$$P_T^* = P_{ave}^* + K_T S^* \tag{7}$$

$$P_{ave}^* = \frac{1}{n}\sum_{i=1}^{n} p_i^* \tag{8}$$

$$S^* = \left[ \frac{1}{n-1} \sum_{i=1}^{n} (p^* - p^*_{ave})^2 \right]^{\frac{1}{2}} \tag{9}$$

where $P^*_T$, $P^*_{ave}$, and $S^*$ are as defined previously but based on the logarithmically transformed $P_i$ values, i.e., $P^*$ of Equation (6). $K_T$ is the Pearson frequency factor which depends on return period ($T$) and skewness coefficient ($C_s$).

The skewness coefficient, $C_s$, is required to compute the frequency factor for this distribution. The skewness coefficient is computed by Equation (10) [33,34].

$$C_s = \frac{n \sum_{i=1}^{ni} (p^*_i - p^*_{ave})^3}{(n-1)(n-2)(S^*)^3} \tag{10}$$

$K_T$ values can be obtained from tables in many hydrology references [33]. By knowing the skewness coefficient and the recurrence interval, the frequency factor, $K_T$, for the LPT III distribution can be extracted. The antilog of the solution in Equation (7) will provide the estimated extreme value for the given return period.

The goodness-of-fit test is used to determine the best-fitting distribution. The validity of a stated or assumed probability distribution model is tested using goodness-of-fit test statistics [35]. In general, a goodness-of-fit test is used to determine how well the observed data matches the fitted (assumed) model. The Kolmogorov–Smirnov (K-S) test, the root mean square error (*RMSE*) test, and the Chi-square test are the most often used goodness-of-fit tests. To choose the best probability distribution to explain the data, the goodness-of-fit test based on the Kolmogorov–Smirnov (K-S) test is utilized. The average error is estimated using the root mean square error (*RMSE*) [36]. The *RMSE* number denotes the average difference between the expected and actual values and it is commonly used to evaluate the models [37]. The *RMSE* is useful because it displays inaccuracies in the units of the element of interest, which assists in the interpretation of the data, despite the fact that it is widely known that the lower the *RMSE*, the better the model performance. However, Singh et al. [38] and Moriasi et al. [39] presented criteria to define what is considered a low *RMSE* based on the standard deviation (SD) of the data. Close to zero *RMSE* readings imply a great match. However, values less than half the SD of the data may be considered low. *RMSE* was given by:

$$RMSE = \sqrt{\frac{1}{n} \sum_{i=1}^{n} \left[ R_{obs} - R_{exp} \right]^2} \tag{11}$$

where,

$R_{obs}$ is the total observed rainfall depth,
$R_{exp}$ is the expected total rainfall depth from the probability distribution, and $n$ is the number of data points at the station.

## 4. Results and Discussion

The morphometric analysis of Wadi Al-Lith is primarily based on the processing, analysis, and tracing of the drainage network using the ASTER Digital Elevation Model with 30 m resolution. The different morphometric parameters are calculated using the WMS 11.0 and Arc Hydro10.3 package (Table 1). Wadi Al-Lith is an elongated drainage basin with a length of 109 km and maximum width of 50 km.

The maximum rain depths of the three metrological stations were investigated for different time periods using the Gumble and log-Pearson type III distributions, and these depths are presented in Figures 2–4. According to data from available stations, the most precipitation falls during the winter and autumn seasons, particularly in the months of January, November, and December. It shows that the greatest amount of rainfall was in 1981, where it reached >120 mm in station TA233, >140 mm in 1996 at station J107, and >150 mm at station J108. It can be noted that the different annual rainfall amounts are

different from year to year, but the annual average of rainfall in these stations was nearly between 55 to 65 mm. The statistical parameters of the annual maximum daily precipitation for the three meteorological stations (TA-233, J-107, and J108) are presented in Table 2.

**Table 1.** Morphometric parameters of Wadi Al-Lith catchment area.

| Basin area | 3089 km$^2$ |
| --- | --- |
| Maximum basin slope | 0.52 |
| Average overland flow | 105.6 km |
| Basin lengths | 108.5 km |
| Perimeter | 53 km |
| Mean basin elevation | 898 m |
| Maximum flow distance | 148.2 km |
| Maximum flow slope | 0.02 |
| Centroid stream distance | 75.1 km |
| Centroid stream slope | 0.01 |

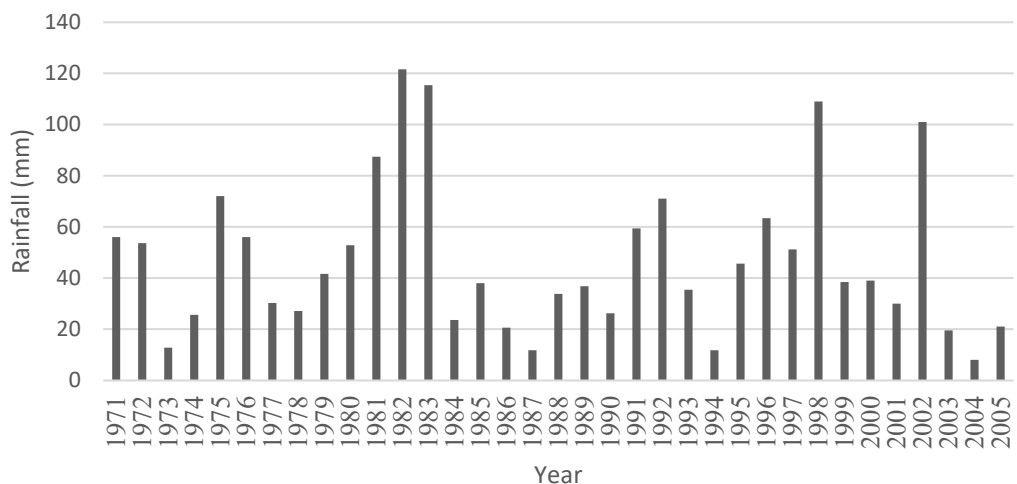

**Figure 2.** Maximum 24 h rainfall precipitation at TA233 station.

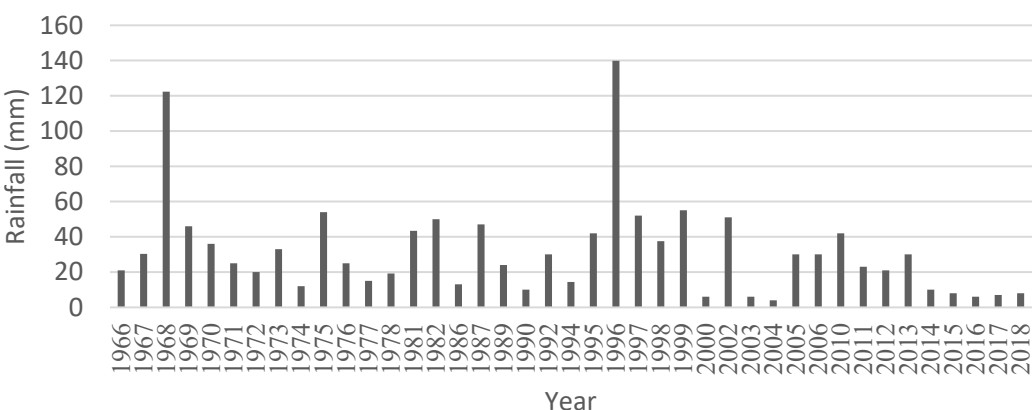

**Figure 3.** Maximum 24 h rainfall precipitation at J107 station.

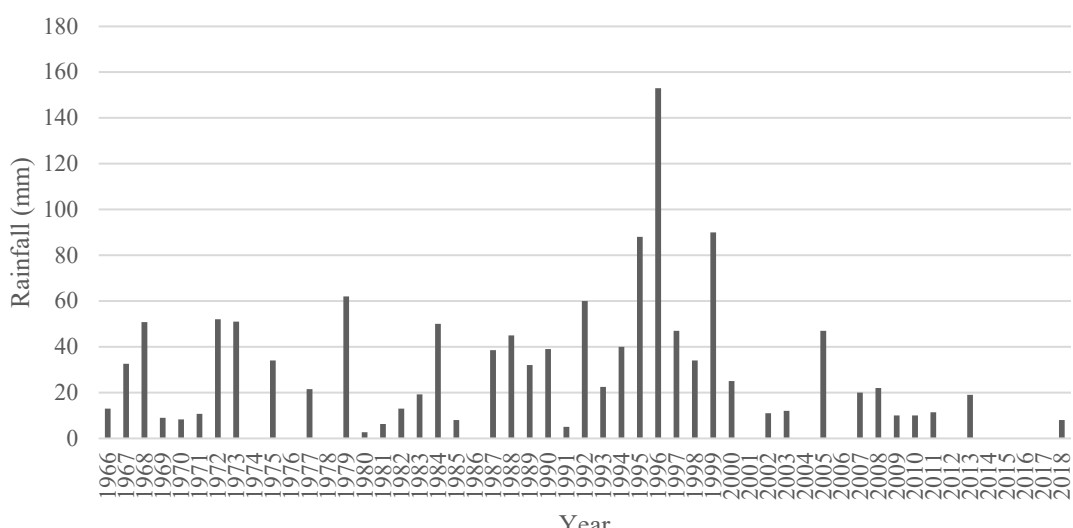

**Figure 4.** Maximum 24 h rainfall precipitation at J108 station.

**Table 2.** Statistical characteristics of the annual maximum daily precipitation of all stations.

| Station | TA-233 | J-107 | J-108 |
|---|---|---|---|
| Sample size | 35 | 41 | 48 |
| Minimum (mm) | 13 | 4 | 1 |
| Maximum (mm) | 122 | 140 | 153 |
| Median (mm) | 39 | 25 | 19.6 |
| Mean (mm) | 49.6 | 31.7 | 28 |
| Standard deviation | 28.8 | 27.6 | 29 |
| Variation coefficient | 0.581 | 0.87 | 1.04 |
| Skewness coefficient | 1.09 | 2.38 | 2.1 |
| Kurtosis coefficient | 3.1 | 8.84 | 8.27 |

The analysis of precipitation frequencies to perform extreme precipitation curves is part of the study of the theoretical statistical distribution that best suits the study area. Different statistical distributions have been tested for the elaboration of IDF curves in the wadi, which are widely used such as the Gumbel, generalized extreme value (GEV), Gamma, exponential, and log-Pearson Type III (LPT III). Apparently, and from the obtained results, Gumbel and log-Pearson Type III best fit the data.

To carefully choose which of the distributions should be used to make the appropriate IDF curves, tests were performed such as the Chi-square goodness of fit, Kolmogorov–Smirnov, and *RMSE*. The Chi-square test compares the observed data with the empirical distribution, and then decide how good a fit between the observed and the expected frequencies obtained from the hypothesized distributions. By making this comparison with the probabilities of each distribution, based on precipitation data recorded, with the same expected period given by the distributions studied, and assuming a significance level of $\alpha = 0.05$, the results obtained can be seen in Table 3. The statistical comparison by Chi-square and K–S tests for goodness of fit showed that the best-fit distribution was log-Pearson Type III for the TA-233 and J-108 stations; lowest values of Chi-square and K-S tests (0.25, 0.088), (9.375, 0.096) were obtained for TA-233 and J-108 stations, respectively. For the J-107 station, the Gumbel distribution has the lowest value of Chi-square and K-S tests which are 3.2 and 0.108, respectively. The overall results and in general the analysis of K–S and Chi-square tests for goodness of fit lead to the conclusion that log-Pearson T-III can be adopted to predict heavy rains. Once the distribution that most closely approximates

reality has been carefully selected, a forecast can be made of the maximum rainfall that can occur for the most usual return periods (T) for this type of study. In this case, the return periods of 2, 5, 10, 25, 50, and 100 years were chosen.

**Table 3.** Summary of the best-fit distribution of stations.

| Stations | Gumbel | | GEV | | Gamma | | Exponential | | LPT III | |
|---|---|---|---|---|---|---|---|---|---|---|
| | Chi-Square | K-S | Chi-Square | K-S | Chi-Square | K-S | Chi-Square | K-S | Chi-Square | K-S |
| TA-233 | 1 | 0.11 | 1 | 0.112 | 2.875 | 0.1 | 16.375 | 0.324 | **0.25** | **0.088** |
| J-108 | 11.125 | 0.141 | 12.375 | 0.113 | 10.875 | 0.1 | 10.875 | 0.111 | **9.375** | **0.096** |
| J-107 | **3.2** | 0.108 | 10.4 | **0.089** | 4.8 | 0.11 | 9.6 | 0.148 | 6 | 0.101 |

Bold font in the table shows the minimum K–S and Chi-square values obtained from the data.

Table 4 displays the root mean square error values in millimeters for various distributions for all stations. The results show that the LPT III distribution is the best, which is supported by the K–S and Chi-square tests. Such a result agrees with the previous results obtained on KSA as the results obtained by Abdeen [23] who noticed that LPT III is the best. In addition, results were obtained by AL-Husson [20] in developing the IDF for Riyadh region, Saudi Arabia. The values of SD are 28.8 mm, 27.6 mm, and 29 mm for TA233, J-107, and J-108 stations, respectively. As shown in Table 4, the *RMSE* values of all tested distributions ranged between 4.19–8.387 mm, 6.315–9.884 mm and 4.486–9.132 mm for Ta233, J-107, and J-108 stations, respectively. On the other hand, the *RMSE* values in the case of LPT III ranged between 4.19 to 6.315 which are less than the other distributions. When those values of *RMSE* are compared with SD values, it is noted that all values of *RMSE* are less than half of the SD values mentioned in Table 2. Therefore, the *RMSE* values are considered low and acceptable.

**Table 4.** Root mean square error in mm for different distribution.

| | Gumbel | GEV | Gamma | Exponential | LP III |
|---|---|---|---|---|---|
| **Station** | *RMSE* | *RMSE* | *RMSE* | *RMSE* | *RMSE* |
| **TA-233** | 4.8 | 6.722 | 4.719 | 8.387 | **4.19** |
| **J-108** | 8.359 | 9.132 | 5.581 | 5.847 | **4.486** |
| **J-107** | 8.864 | 7.159 | 9.884 | 7.714 | **6.315** |

Bold font in the table shows the minimum *RMSE*.

The maximum of 24 hr rainfall depth for all stations at six return periods is estimated as shown in Table 5. Figure 5a–f shows the probabilities of exceedance using the two distributions, Gumbel and log-Pearson type III, for data from three stations located upstream and downstream of Wadi Al-Lith.

**Table 5.** Maximum 24 hr rainfall depth (mm) for 6 return periods.

| | Return Period (Year) | 2 | 5 | 10 | 25 | 50 | 100 |
|---|---|---|---|---|---|---|---|
| **J107 station** | Gumbel (mm) | 27.1 | 46.1 | 58.7 | 74.6 | 86.5 | 98.2 |
| | Log-Pearson type III | 23.6 | 45.9 | 64.2 | 91.2 | 114 | 139 |
| **J108 station** | Gumbel (mm) | 27.4 | 48.1 | 61.8 | 79.1 | 92 | 105 |
| | Log-Pearson type III | 23.3 | 47.3 | 67.7 | 98.3 | 124 | 153 |
| **TA233 station** | Gumbel (mm) | 41.7 | 66.6 | 83.1 | 104 | 119 | 135 |
| | Log-Pearson type III | 39.5 | 67.3 | 87.4 | 114 | 134 | 155 |

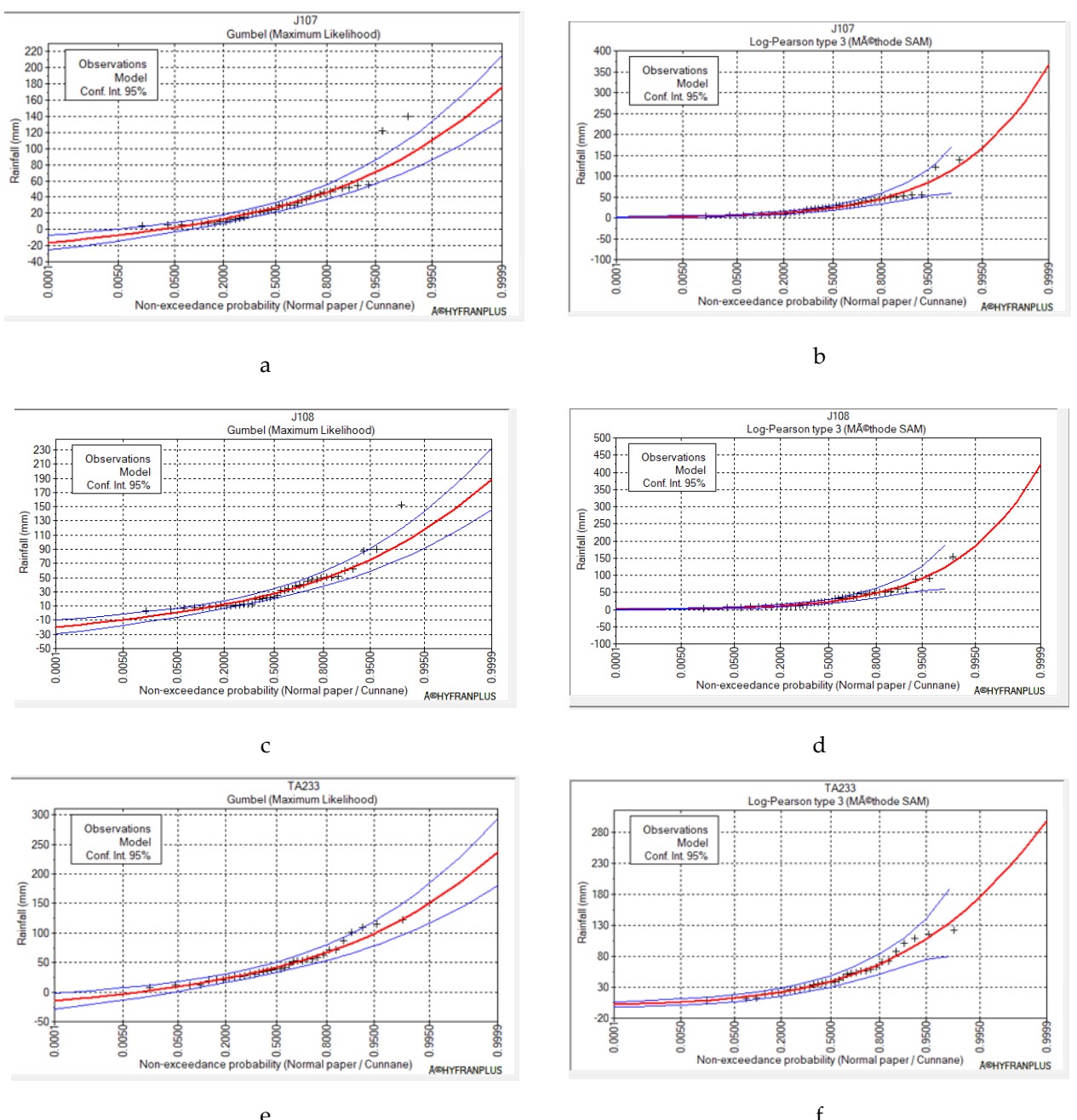

**Figure 5.** (**a**–**f**) Gumbel and log-Pearson type III fits of maximum daily RF of J-107, J-108, and TA-233.

It has been demonstrated that there are minor differences between the findings obtained using the two distributions, with the log-Pearson type III method providing slightly better results than Gumbel's. Using the Thiessen polygon, the Wadi Al-Lith region was divided into three regions (A, B, and C) based on the area covered from the station, as shown in Figure 6. The IDF curves in Figures 7–9 show that rainfall estimates increase as return periods increase, and rainfall intensities decrease with rainfall duration for all return periods. It is seen in Figure 5a–f and the obtained results from the Chi-square goodness-of-fit, Kolmogorov–Smirnov, and *RMSE* tests that IDF curves are developed for this region based on log-Pearson type III because they are more suitable and best fit the data.

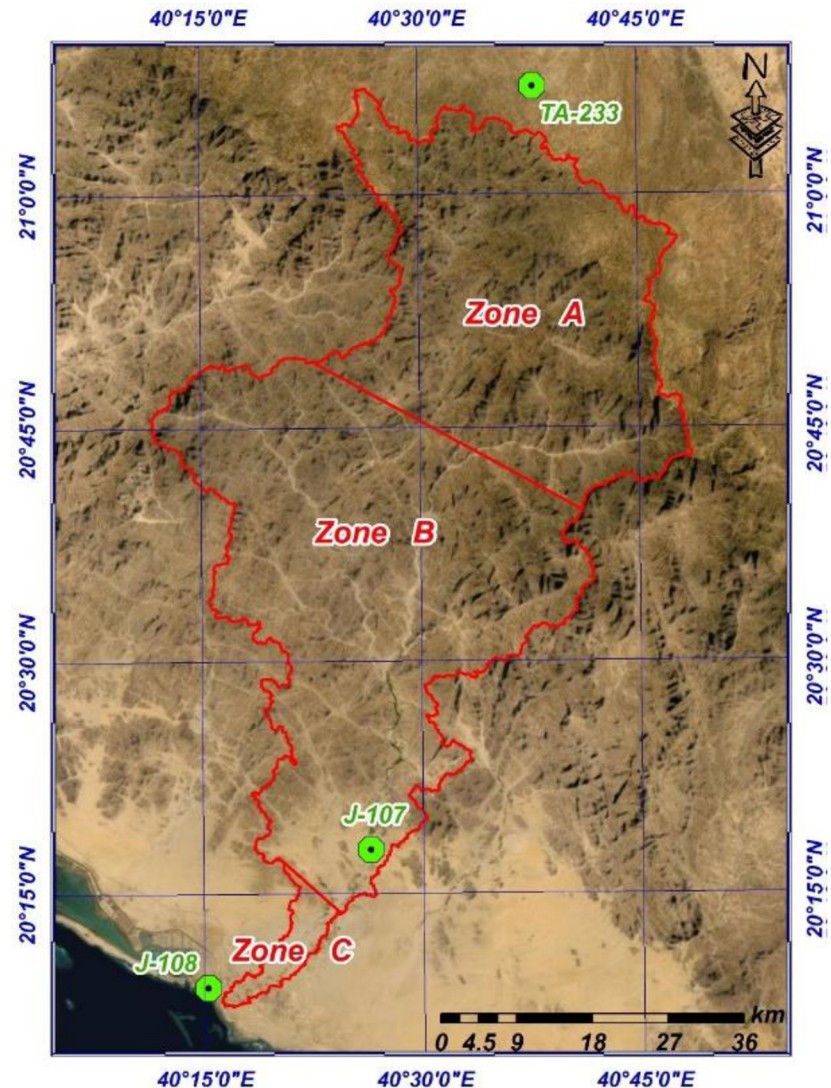

**Figure 6.** Thiessen polygon zones of the TA-233, J-107, and J-108 stations.

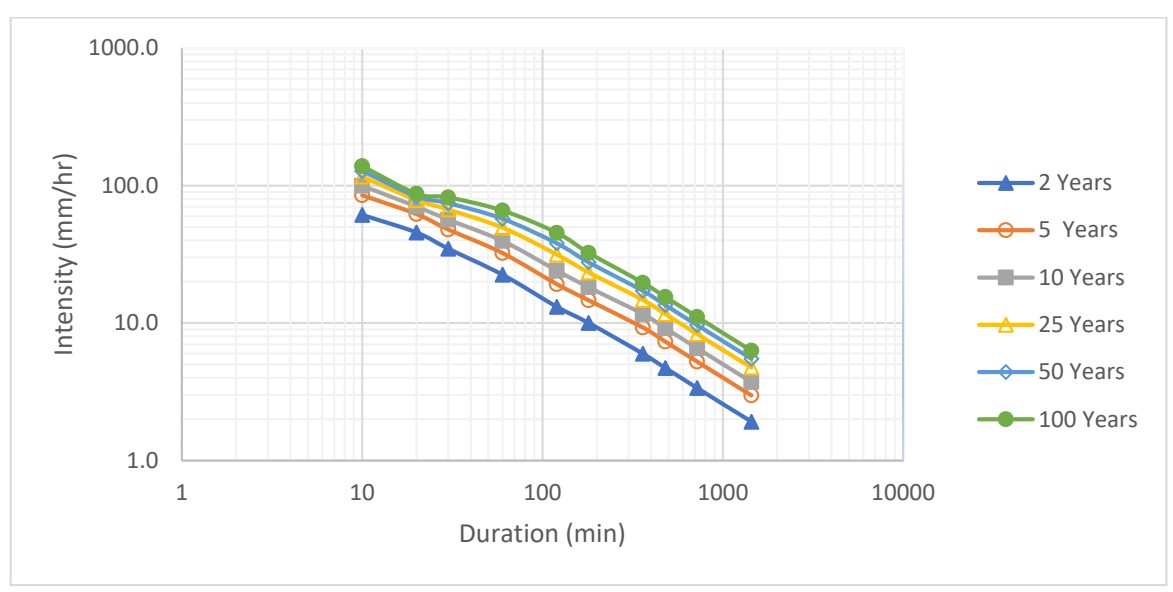

**Figure 7.** IDF curves by log-Pearson type III distribution at Al-Lith (Zone A).

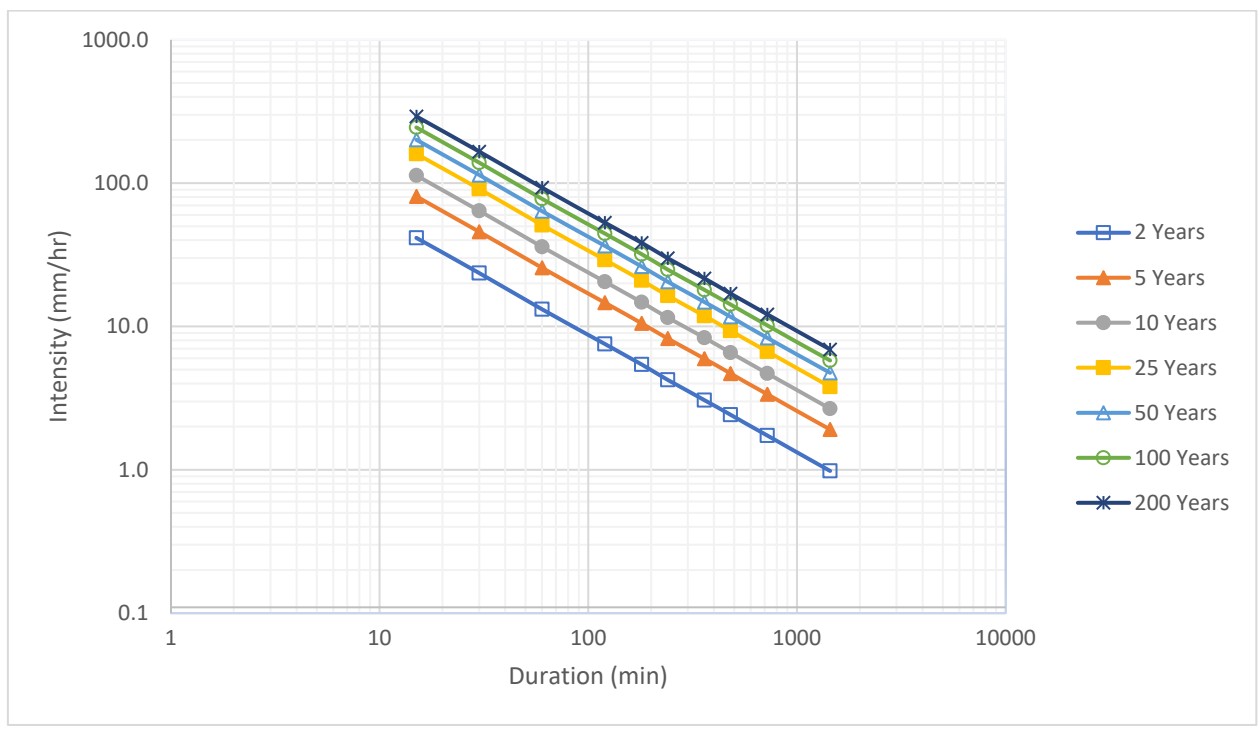

**Figure 8.** IDF curves by log-Pearson type III distribution at Al-Lith (Zone B).

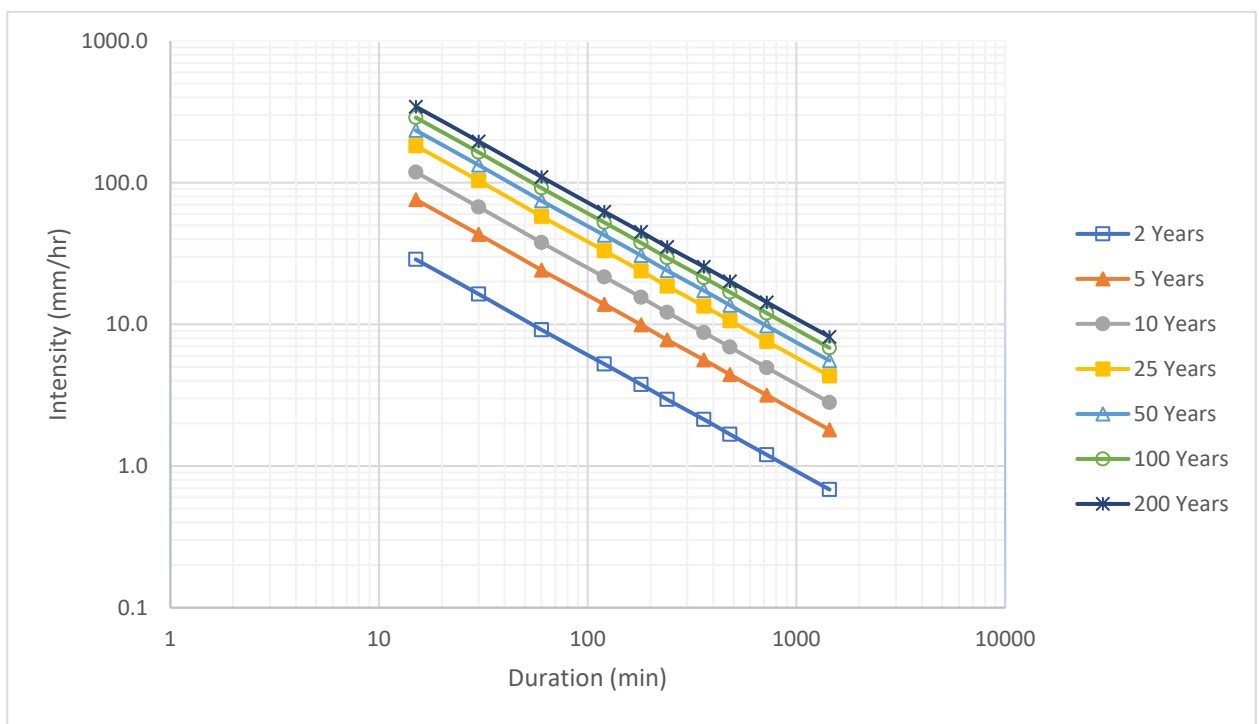

**Figure 9.** IDF curves by log-Pearson type III distribution at Al-Lith (Zone C).

From 1980 to 2018, an isohyetal map based on data from rain gauge stations shows an increase in rainfall on the escarpment ridge to >180 mm (Figure 10). On the other hand, as you travel west, the level drops until it reaches <60 mm in the coastal area. When the annual rainfall amounts for the three rain gauge stations within the Wadi Al-Lith catchment area (TA233, J107, and J108) are compared, it is clear that the annual rainfall amount in the northeastern part of Wadi Al-Lith, which is characterized by high altitude, reaches

>180 mm, <120 mm in central part, and then drops to <50 mm in the coastal area (Figure 10). The results of the research for the three stations with different theoretical distributions clearly show an increase in the maximum 24 hr rainfall amount in the northeastern part of the wadi (Region A), as can be seen in Table 5, where the maximum 24 h rainfall depth values expected for the return periods under study can be seen. The maximum 24 hr rainfall amount drops until it reaches 50% less in the coastal area (Region C). The descriptive statistics of the two regions (B and C), which are close to each other and are also close to sea level, for the same return periods, show homogeneity of results. In comparison between the two regions B and C at the same return period, it is noted that there are no big variations between the obtained results, where the maximum 24 hr rainfall amount in region B of the wadi are little bit higher than in region C, with a range of 7% at (25 years return period) to 10% (at 100 years return period).

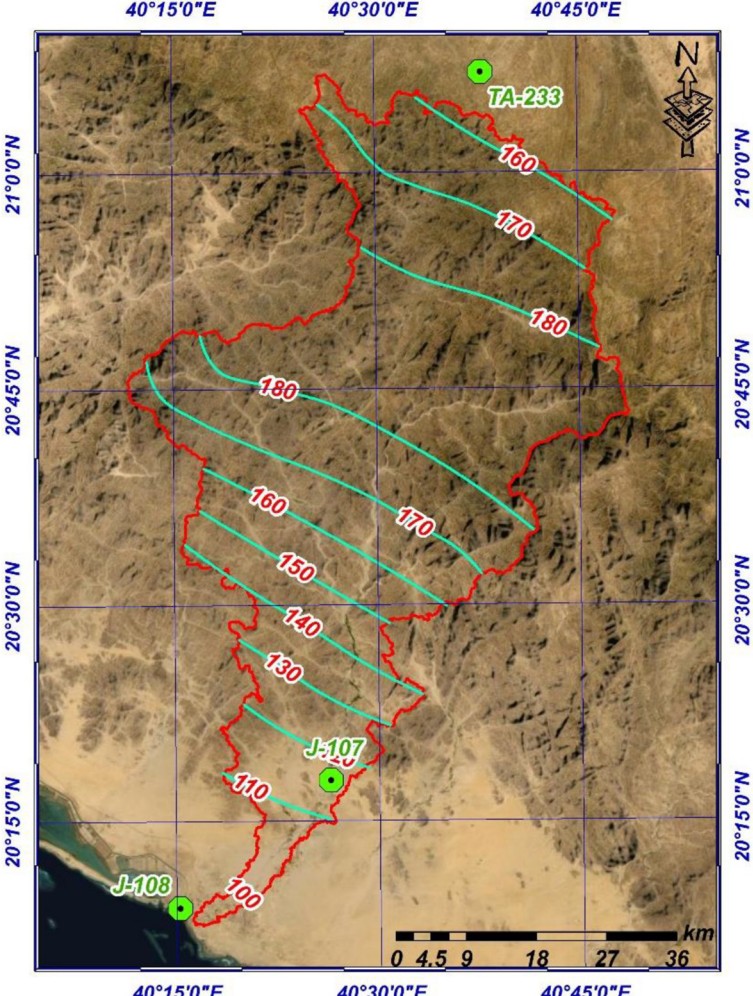

**Figure 10.** Spatial distribution of maximum 24 h rainfall mm (annual basis).

## 5. Conclusions

Intensity–duration–frequency (IDF) curves are valuable knowledge resources for hydrologists and engineers working on water project design. The most precipitation comes during the winter and autumn seasons, according to data from available stations. Historical rainfall intensity records from three major meteorological stations in the Lith area were created using the best approaches to distribution of probability functions. It was demonstrated that the findings obtained using the two distributions vary significantly, with the log-Pearson type III approach producing slightly better results than Gumbel's. The rainfall amount in the northeastern part of Wadi Al-Lith, which is characterized by high

altitude, reaches >180 mm. In the central part, rainfall drops to <120 mm and 50 mm in the coastal area. There is an increase in the maximum 24 hr rainfall amount in the northeastern part of the wadi (region A), where it drops until it reaches 50% less in the coastal area (region C). In comparison between the two regions B and C, it is noted that there are no big variations between the obtained results, where the maximum 24 hr rainfall amount in region B of the wadi are little bit higher than in region C, with a range of 7% at (25 years return period) to 10% (at 100 years return period). The isohyetal map and IDF curves show that the rainfall amount is a function of altitude, and the orographic effect of the escarpment ridge on rainfall increase is clearly visible in the northeastern region.

Further research is recommended using satellite production, i.e., the Tropical Rainfall Measuring Mission (TRMM), to investigate and verify the reliability of the observed rainfall data, as well as overcome the problem of limited rainfall station numbers in the catchment area. Therefore, the obtained IDF curves can be modified later after adding the satellite production, which will also help development of the curves on the short duration basis. The adequate and reliable estimation of IDF curves will allow and guarantee the planning and optimal design of hydraulic structures, as well as future flood hazard studies, and consequently, disaster risk management will also benefit directly from that study.

**Author Contributions:** Conceptualization, I.H.E.; methodology, A.Q.K.; software, M.E.A. and A.Q.K.; validation, I.H.E.; formal analysis, M.E.A.; investigation, I.H.E.; resources, A.Q.K.; data curation, A.Q.K.; writing—original draft preparation, A.Q.K.; writing—review and editing, I.H.E. and M.E.A.; visualization, I.H.E.; supervision, I.H.E.; project administration, I.H.E.; funding acquisition, I.H.E. All authors have read and agreed to the published version of the manuscript.

**Funding:** This research was funded by the National Plan for Science, Technology and Innovation (MAARIFAH), King Abdulaziz City for Science and Technology, Kingdom of Saudi Arabia, Award Number (13-WAT1027-02).

**Institutional Review Board Statement:** Not applicable.

**Informed Consent Statement:** Not applicable.

**Acknowledgments:** The authors extend their sincere appreciations to the National Plan for Science, Technology and Innovation (NPST)-King Saud University for funding the Research Project (13-WAT1027-02).

**Conflicts of Interest:** The authors declare no conflict of interest.

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
