# Peer review of "Spatiotemporal Variability of Intensity–Duration–Frequency (IDF) Curves in Arid Areas: Wadi AL-Lith, Saudi Arabia as a Case Study"

_hydrology, doi:10.3390/hydrology9010006_

Round 1

Reviewer 1 Report

The topic of the paper under the title “Spatio-temporal variability of Intensity–Duration–Frequency (IDF) curves in arid areas: Wadi AL-Lith, Saudi Arabia as a case study” is interesting and within the scope of the Hydrology journal. However, some issues require clarification and improvement. Therefore, I recommend major revisions before the publication of the manuscript. Please find some details below.

  1. The Introduction should be expanded and a broader literature review carried out. Currently, the list of References is quite poor and includes mainly items older than 5 years. Please analyze the literature from recent years and in relation to this literature, emphasize the originality of the research described in the manuscript. Please also remember to avoid using lumped references. For this reason, the sentence from lines 41-43 should also be brightened up.
  2. Section 3 is described in a rather chaotic way. The Equations and their numbers should be aligned according to the mdpi guidelines. In line 144 the authors refer to subsection 3.1, which is missing from the manuscript. There is also a problem with indexes. For example, in line 113, "PT should be changed to" PT". Please check the entire section entitled" Methodology and data collection" carefully to eliminate mistakes. In the opinion of this reviewer, it is also advisable to add a list of abbreviations.
  3. The description in section 4 is very poor. There are many figures and tables in this section, but their description is kept to a minimum. There is also no discussion of the results in relation to the achievements of other authors.
  4. The Conclusions section should be supplemented with directions for further research. It should also describe for whom the results presented in the manuscript may be useful and why.

Please also consider the following issues:

  • Please change “Type of the Paper (Article.)” (line 1) to “Article”.
  • Keywords: Please change “IDF curves” to “Intensity-Duration-Frequency (IDF) curves”
  • Figures 2a-2c should have one common title. Otherwise, they should be given consecutive numbers. The same is true for Figures 5a-5c.
  • Figure 3a-3f – Individual figures should be described with letters a-f. The title should describe what each of them represents.

Best regards

Author Response

Dear Editor-in-Chief;

Thank you for your e-mail informing us about our manuscript status.  Please find below our replies to the viewer’ comments.

First of all, the authors would like to express their thanks and appreciation for the reviewer for the constructive and valuable comments. Below is our reply to each of the comments.

Reviewer #1:

Reviewer’s comment:

The Introduction should be expanded and a broader literature review carried out. Currently, the list of References is quite poor and includes mainly items older than 5 years. Please analyze the literature from recent years and in relation to this literature, emphasize the originality of the research described in the manuscript. Please also remember to avoid using lumped references.

Author response:

The introduction is expanded and references are added in recent years.

The originality of the research is described on page 2 lines 68-73.

Reviewer’s comment:

For this reason, the sentence from lines 41-43 should also be brightened up.

Authors’ response:

It is done see page 1 Line 34-35

Reviewer’s comment:

Section 3 is described in a rather chaotic way. The Equations and their numbers should be aligned according to the dpi guidelines. In line 144 the authors refer to subsection 3.1, which is missing from the manuscript. There is also a problem with indexes. For example, in line 113, "PT should be changed to" PT". Please check the entire section entitled" Methodology and data collection" carefully to eliminate mistakes. In the opinion of this reviewer, it is also advisable to add a list of abbreviations.

Authors’ response:

Thank you for this comment, there were mistakes in that section and corrected. See page 4-5  

Reviewer’s comment:

The description in section 4 is very poor. There are many figures and tables in this section, but their description is kept to a minimum. There is also no discussion of the results in relation to the achievements of other authors.

Authors’ response:

The description was expanded by adding more discussions on the obtained results. See pages 7-8, line 187-222 also page 12, line 249-257.

Reviewer’s comment:

The Conclusions section should be supplemented with directions for further research. It should also describe for whom the results presented in the manuscript may be useful and why.

Authors’ response:

Further research with some directions was added. See page 13 lines 275-281.

 Thank you for that comment, we described for whom the results presented in the manuscript. See page 13 line 269-273.

Reviewer’s comment:

  • Please change “Type of the Paper (Article.)” (line 1) to “Article”.

Authors’ response:

It is done (see page 1, line 1).

 Reviewer’s comment:

  • Keywords: Please change “IDF curves” to “Intensity-Duration-Frequency (IDF) curves”

Authors’ response:

 It is done (see page 1, line 19).

Reviewer’s comment:

  • Figures 2a-2c should have one common title. Otherwise, they should be given consecutive numbers. The same is true for Figures 5a-5c.

Authors’ response:

Figures 2a-2c changed to figures 2-4. See pages 6-7

In figures, 5a-5a changed to figures 7-9. See pages 11-12

Reviewer’s comment:

  • Figure 3a-3f – Individual figures should be described with letters a-f. The title should describe what each of them represents.

Authors’ response:

It is done, figure 3 became figure 5(see page 9).

Thank you.

Reviewer 2 Report

Dear Editor.

I have finished my review on the proposed paper “Spatio-temporal variability of Intensity–Duration–Frequency (IDF) curves in arid areas: Wadi AL-Lith, Saudi Arabia as a case 3 study”, hydrology-1503719-peer-review-v1.

Summary of the manuscript:

In the proposed paper, the authors goals are to evaluate two extreme distributions (Gumbel and Log- Pearson type III), in order to build the IDF curves of the study area and provide a map of the spatiotemporal distribution of 24h max rainfalls.

General review:

  1. The English language used in the paper is good, but needs to be improved and the whole paper should be checked by a native speaker, since several grammatical and syntactical errors were noticed.
  2. The Introduction is very weak. There are very few references that provide respective studies from other countries or/and other studies that deal with extreme rainfall evaluation. More literature should be added in the Introduction. Below I give some papers to add and begin with. At the end of Introduction, authors do not state in a clear way the goals of the research and they do not indicate the novelty of the paper (if there is any).
  3. The section of the methodology of the research is very low, and significant parts are missing. Below I give specific comments.
  4. The results I think that are interesting, but the authors do not underline the significance of the paper (if there is any). In my opinion tables and figures are easily understandable, however some changes should be done.
  5. The Discussion of the paper is poor. Basically, there is no Discussion in the “4. Results and discussion” section. It was expected to see some comparisons with other studies from Europe or other countries that applied similar research methodology. Are there other studies to compare your findings?

Points for revision:

Lines 26-27: How do you know about “Precipitation patterns and extremes”? You should add literature here.

Line 58: You refer here the Root Mean Square Error (RMSE). But I didn’t see the results of RMSE in the text. Where are the results of RMSE?

Figure 1: Add some points with the cities of the are and the meteorological stations.

Methodology and data collection: In this section you should refer that you use the RMSE and provide the equation.

Lines 102-103: Please, add literature.

Lines 107-109: you should provide the descriptive statistics of the rainfall data for each station. See this paper for example (Kastridis and Stathis 2017). You should add tables with these statistics.

Lines 109, 113 and elsewhere: You have wronged the name of Gumbel (Gamble??).

You have to provide literature for the two distributions that you use.

You have pre-chosen two distributions in order to build the IDF curves. How do you know that these two distributions have the best fit to your data??? There are other distributions (EV max-1, EV max-2, GEV max, Lognormal, L-moments…. Singh 1998, Morrison and Smith 2002) that maybe fit better to your rainfall data. You should check and other distributions if you want your results to be acceptable and reliable. You should the proposed literature and discuss why you chose the specific distributions.

Also, (and very significant) how you checked the good performance and fit to the data of the two distributions? Did you use any statistical index to check it? In these studies, the x2 and Kolmogorov–Smirnov tests are applied to find which distribution has the best fit to the data. I see in your paper that you didn’t use anything of these tests. You should add these tests in your paper, and provide literature and compare your results with other papers. Below I give literature to add in your paper and begin your searching (Kastridis et al. 2021). You should this in your paper providing more literature.

The time series of the meteorological stations have different length of years. It is already known that the size (length) of time series significantly affects the selection of the appropriate distribution to be fitted to the rainfall data (Koutsogiannis and Baloutsos 2000, Kastridis and Stathis 2017). You should this in your paper using the providing literature and other papers.

Lines 198-205: This paragraph is the exactly the same with the paragraph in lines 189-196. Be careful. Your paper is not prepared in detail.

Lines 222-225 and figure 6: And here at the end of the paper is a surprise! You talking about the TRMM (I think that you mean about the “Tropical Rainfall Measuring Mission”) annual rainfall without saying anything in your abstract or Methodology section. You should add information about TRMM in methodology and provide literature of other papers that use TRMM.

There is no discussion. You should discuss your findings comparing your results with other relative studies.

I read all your paper but I didn’t find anything about the RMSE that said in Introduction. Here are two papers about the RMSE that you can use and add in your paper

References

Kastridis A., Stathis D. (2017) The Effect of Rainfall Intensity on the Flood Generation of Mountainous Watersheds (Chalkidiki Prefecture, North Greece). In: Karacostas T., Bais A., Nastos P. (eds) Perspectives on Atmospheric Sciences. Springer Atmospheric Sciences. Springer, Cham. https://doi.org/10.1007/978-3-319-35095-0_48

Kastridis, A.; Theodosiou, G.; Fotiadis, G. Investigation of Flood Management and Mitigation Measures in Ungauged NATURA Protected Watersheds. Hydrology 2021, 8, 170. https://doi.org/10.3390/hydrology8040170

Koutsoyiannis D, Baloutsos G (2000) Analysis of a long record of annual maximum rainfall in Athens, Greece, and design rainfall inferences. Nat Hazards 29:29–48

Moriasi, D.; Arnold, J.; Van Liew, M.; Bingner, R.; Harmel, R.; Veith, T. Model evaluation guidelines for systematic quantification of accuracy in watershed simulations. Trans. ASABE 2007, 50, 885–900.

Singh, J.; Knapp, H.V.; Arnold, J.; Demissie, M. Hydrological modeling of the Iroquois river watershed using hspf and swat1. J. Am. Water Resour. Assoc. 2005, 41, 343–360.

Singh VP (1998) Extreme value type 1 distribution. Entropy-based parameter estimation in hydrology. Springer Netherlands, pp 108–136. doi:10.1007/978-94-017-1431-0_8

Morrison J, Smith J (2002) Stochastic modeling of flood peaks using the generalized extreme value distribution. Water Resour Res 38(12):1305. doi:10.1029/2001WR000502.

Author Response

Dear Editor-in-Chief;

Thank you for your e-mail informing us about our manuscript status.  Please find below our replies to the viewer comments.

First of all, the authors would like to express their thanks and appreciation to the reviewer for the constructive and valuable comments. Below is our reply to each of the comments.

Reviewer #2:

Reviewer’s comment:

The English language used in the paper is good but needs to be improved and the whole paper should be checked by a native speaker since several grammatical and syntactical errors were noticed.

Authors’ response:

Grammatical and other errors were checked.

Reviewer’s comment:

The Introduction is very weak. There are very few references that provide respective studies from other countries or/and other studies that deal with extreme rainfall evaluation. More literature should be added to the Introduction. Below I give some papers to add and begin with. At the end of the Introduction, the authors do not state in a clear way the goals of the research, and they do not indicate the novelty of the paper (if there is any).

Authors’ response:

The introduction is expanded and references are added in recent years. See page 2 lines 55-66.

The originality of the research is described on page 2 lines 68-73.

Reviewer’s comment:

The section of the methodology of the research is very low, and significant parts are missing. Below I give specific comments.

Authors’ response:

It was expanded and a description of the goodness of fit tests was added. See page 4 line 144-155

Reviewer’s comment:

The results I think are interesting, but the authors do not underline the significance of the paper (if there is any). In my opinion tables and figures are easily understandable, however, some changes should be done.

Authors’ response:

The description of figures was expanded and more discussion was added. See page 12 line 249-257.

Reviewer’s comment:

The Discussion of the paper is poor. Basically, there is no Discussion in the “4. Results and discussion” section. It was expected to see some comparisons with other studies from Europe or other countries that applied similar research methodology. Are there other studies to compare your findings?

Authors’ response:

The discussion was expanded and the results were compared to other studies.  See pages 7-8, lines 187-222.

Reviewer’s comment:

Lines 26-27: How do you know about “Precipitation patterns and extremes”? You should add literature here.

Authors’ response:

It is done. See page1 line 24

Reviewer’s comment:

Line 58: You refer here to the Root Mean Square Error (RMSE). But I didn’t see the results of RMSE in the text. Where are the results of RMSE?

Authors’ response:

The results of RMSE were added. See page 8 table 4

Reviewer’s comment:

Figure 1: Add some points with the cities of the area and the meteorological stations.

Authors’ response:

It is done. See page 3 Figure 1

Reviewer’s comment:

Methodology and data collection: In this section you should refer that you use the RMSE and provide the equation.

Authors’ response:

The equation of RMSE and brief description of good fitness were added. See page 4 line 144-155.

Reviewer’s comment:

Lines 102-103: Please, add literature.

Authors’ response:

It is done. See page3 line 101

Reviewer’s comment:

Lines 107-109: you should provide the descriptive statistics of the rainfall data for each station. See this paper for example (Kastridis and Stathis 2017). You should add tables with these statistics.

Authors’ response:

It is done. See page5 table 2

Reviewer’s comment:

Lines 109, 113 and elsewhere: You have wronged the name of Gumbel (Gamble??).

Authors’ response:

It is corrected in the all manuscript. See page 3 line 109

Reviewer’s comment:

You have to provide literature for the two distributions that you use.

Authors’ response

It is done. See page 3 line 109 and page 4 line 130

Reviewer’s comment:

You have pre-chosen two distributions in order to build the IDF curves. How do you know that these two distributions have the best fit to your data??? There are other distributions (EV max-1, EV max-2, GEV max, Lognormal, L-moments…. Singh 1998, Morrison and Smith 2002) that maybe fit better to your rainfall data. You should check and other distributions if you want your results to be acceptable and reliable. You should the proposed literature and discuss why you chose the specific distributions.

Authors’ response

In fact, we started with the well-known distributions, and after doing the frequency analysis we could reach two of them that best fit the data.  To choose carefully which of the distributions should be used to make the appropriate IDF curves, tests were performed such as the Chi-square Goodness of Fit, Kolmogorov-Smirnov, and RMSE. This part is explained in details with the obtained results and the added Tables (3&4). See page 7, 8.

Reviewer’s comment:

Also, (and very significant) how you checked the good performance and fit to the data of the two distributions? Did you use any statistical index to check it? In these studies, the x2 and Kolmogorov–Smirnov tests are applied to find which distribution has the best fit to the data. I see in your paper that you didn’t use anything of these tests. You should add these tests in your paper, and provide literature and compare your results with other papers. Below I give literature to add in your paper and begin your searching (Kastridis et al. 2021). You should this in your paper providing more literature.

Authors’ response

Chi-square Goodness, Kolmogorov-Smirnov, RMSE and statistical parameters are added and described in page 7-8   

Reviewer’s comment:

The time series of the meteorological stations have different length of years. It is already known that the size (length) of time series significantly affects the selection of the appropriate distribution to be fitted to the rainfall data (Koutsogiannis and Baloutsos 2000, Kastridis and Stathis 2017). You should this in your paper using the providing literature and other papers.

Authors’ response

It is done. See page 2 line 55-66

Reviewer’s comment:

Lines 198-205: This paragraph is the exactly the same with the paragraph in lines 189-196. Be careful. Your paper is not prepared in detail.

Authors’ response

Yes, there was a mistake. It is corrected.

Reviewer’s comment:

Lines 222-225 and figure 6: And here at the end of the paper is a surprise! You talking about the TRMM (I think that you mean about the “Tropical Rainfall Measuring Mission”) annual rainfall without saying anything in your abstract or Methodology section. You should add information about TRMM in methodology and provide literature of other papers that use TRMM.

Authors’ response

Yes, we started a further study to use the Tropical Rainfall Measuring Mission (TRMM) in that research to investigate and verify the reliability of the observed rainfall data, as well as overcome the problem of limited rainfall station number in the catchment area, as well as to help for development the curves on the short duration basis. But this needs much work and more investigation of the satellite before we include it to the current research. So, it will be added as further research to develop also the IDF curves on the short duration basis (sub-daily).

Reviewer’s comment:

There is no discussion. You should discuss your findings comparing your results with other relative studies.

Authors’ response

It is done. See page 8 line 212-215

Reviewer’s comment:

I read all your paper but I didn’t find anything about the RMSE that said in Introduction. Here are two papers about the RMSE that you can use and add in your paper

Authors’ response

It is added. See page 4 line 149-155 and page 8 line 212-222

 Thank you

Round 2

Reviewer 1 Report

In my opinion, the paper can be published.

Author Response

The authors would like to express their thanks and appreciation for the reviewer for the constructive and valuable comments.

Reviewer 2 Report

Dear Editor.

The authors responded to most of my comments and suggestions. The paper is significantly improved. However, I have one final suggestion to be added in the paper. 

You have to add some discussion about the results of RMSE. Before the formula of RMSE (lines 147-148) you should give the acceptable values of RMSE (values close to zero (0) indicate best fitting).

Additionally, RMSE describes the difference between model simulations and observations in the units of the variable. RMSE optimal value is zero (0). However, the statement “optimal value is close to zero” is relative and the RMSE alone is not very informative. For example, when the RMSE of a hydrological model in a river (with discharge values ranging between 200-600 m3/sec) is 2.292, could be considered very low and close to zero. However, in another case, for example temperature simulation model (with temperatures values ranging between 0.3 and 5.7 oC), an RMSE of 2.292 is huge.

So, the best way to interpret the RMSE and understand if is acceptable for model evaluation, is to provide the Standard Deviation (SD). It is known from previous studies that RMSE values less than half the SD of the measured data may be considered low and acceptable for model evaluation (Sapountzis et al. 2021, Moriasi et al. 2007, Singh et al. 2005). Compare the values of SD (table 2) with the RMSE values (table 4), discuss in the text the above mentioned, and explain if the RMSE is acceptable in terms of SD. Also add the proposed literature to support your results. (I see that RMSE values are less than the half of SD, so are acceptable).

References.

Moriasi, D., et al. (2007). Model evaluation guidelines for systematic quantification of accuracy in watershed simulations. Transactions of the ASABE, 50(3), 885–900. https://doi.org/10.13031/2013.23153.

Sapountzis M. et al. 2021. Utilization and uncertainties of satellite precipitation data in flash flood hydrological analysis in ungauged watersheds. Global Nest Journal. https://doi.org/10.30955/gnj.003905.

Singh, J., et al. (2005). Hydrological modeling of the iroquois river watershed using hspf and swat1. Journal of the American Water Resources Association, 41, 343–360. https://doi.org/10.1111/j.1752-1688.2005.tb03740.x.

Author Response

First of all, the authors would like to express their thanks and appreciation for the reviewer for the constructive and valuable comments. Below is our reply to the comment.

Reviewer #2:

Reviewer’s comment:
You have to add some discussion about the results of RMSE. Before the formula of RMSE (lines 147-148) you should give the acceptable values of RMSE (values close to zero (0) indicate best fitting).
Additionally, RMSE describes the difference between model simulations and observations in the units of the variable. RMSE optimal value is zero (0). However, the statement “optimal value is close to zero” is relative and the RMSE alone is not very informative. For example, when the RMSE of a hydrological model in a river (with discharge values ranging between 200-600 m3/sec) is 2.292, could be considered very low and close to zero. However, in another case, for example temperature simulation model (with temperatures values ranging between 0.3 and 5.7 oC), an RMSE of 2.292 is huge.
So, the best way to interpret the RMSE and understand if is acceptable for model evaluation, is to provide the Standard Deviation (SD). It is known from previous studies that RMSE values less than half the SD of the measured data may be considered low and acceptable for model evaluation (Sapountzis et al. 2021, Moriasi et al. 2007, Singh et al. 2005). Compare the values of SD (table 2) with the RMSE values (table 4), discuss in the text the above mentioned, and explain if the RMSE is acceptable in terms of SD. Also add the proposed literature to support your results. (I see that RMSE values are less than the half of SD, so are acceptable).

Author response:

It is done, see page 4 lines 150 -155 and page 8 lines 220-226.

Thank You
